# Multimodal Clinical Benchmark for Emergency Care (MC-BEC): A Comprehensive Benchmark for Evaluating Foundation Models in Emergency Medicine

**Emma Chen**[1,4]*    **Aman Kansal**[2]*    **Julie Chen**[2]    **Boyang Tom Jin**[2]
**Julia Rachel Reisler**[2]    **David A Kim**[3]†    **Pranav Rajpurkar**[1]†

[1]Department of Biomedical Informatics, Harvard Medical School
[2]Department of Computer Science, Stanford University
[3]Department of Emergency Medicine, Stanford University School of Medicine
[4]Harvard John A. Paulson School of Engineering and Applied Sciences, Harvard University
`yingchen@g.harvard.edu`
`{amkansal, jchen80, tomjin, jreisler, davidak}@stanford.edu`

## Abstract

We propose the Multimodal Clinical Benchmark for Emergency Care (MC-BEC), a comprehensive benchmark for evaluating foundation models in Emergency Medicine using a dataset of 100K+ continuously monitored Emergency Department visits from 2020-2022. MC-BEC focuses on clinically relevant prediction tasks at timescales from minutes to days, including predicting patient decompensation, disposition, and emergency department (ED) revisit, and includes a standardized evaluation framework with train-test splits and evaluation metrics. The multimodal dataset includes a wide range of detailed clinical data, including triage information, prior diagnoses and medications, continuously measured vital signs, electrocardiogram and photoplethysmograph waveforms, orders placed and medications administered throughout the visit, free-text reports of imaging studies, and information on ED diagnosis, disposition, and subsequent revisits. We provide performance baselines for each prediction task to enable the evaluation of multimodal, multitask models. We believe that MC-BEC will encourage researchers to develop more effective, generalizable, and accessible foundation models for multimodal clinical data.

## 1   Introduction

Emergency Medicine is a critical area of healthcare in which timely and accurate decisions, drawing appropriately on a wide variety of data sources, have a significant impact on patient outcomes [1]. However, developing effective foundation models for electronic health record (EHR) data in Emergency Medicine requires addressing several challenges. EHR data in Emergency Medicine is heterogeneous, including clinical notes, orders, lab results, imaging studies, and physiological waveforms. This heterogeneity can make it difficult to extract meaningful features from the data and integrate them into a single model. Data quality and missingness can also be a significant issue, due to the fast-paced and high-pressure nature of emergency care. Inaccurate or incomplete data can limit the reliability of model results. Clinical interpretability is also critical. Clinicians must be able to understand the model's predictions to make informed decisions. Therefore, models developed for EHR analysis in Emergency Medicine must be transparent and explainable. Finally, the limited scope and standardization of existing datasets are significant challenges to developing foundation models for Emergency Medicine. Many datasets focus on specific patient populations, such as trauma

37th Conference on Neural Information Processing Systems (NeurIPS 2023) Track on Datasets and Benchmarks.

patients, or specific tasks, such as predicting mortality. This orientation to specific subgroups and tasks makes it difficult to compare and evaluate models across tasks and patient populations.

To address these challenges, and to promote the development of robust and clinically useful foundation models for Emergency Medicine, we propose the Multimodal Clinical Benchmark for Emergency Care (MC-BEC), a comprehensive benchmark for evaluating foundation models in Emergency Medicine. MC-BEC is built on a dataset [1] of 102,731 monitored visits made by 63,389 unique patients between 2020 and 2022, and provides a unique opportunity to study acute care in the COVID-19 era. It is the only multimodal medical dataset that exclusively covers patients during this period, while also capturing a wide range of non-COVID pathology. This dataset covers a wide range of information for emergency department (ED) patients, including triage information, prior diagnoses and medications, orders placed in the ED, medication administrations, lab results, continuously monitored vital signs and physiologic waveforms, and free-text reports for radiology studies. With its emphasis on multiple modalities, including continuous waveforms and vital signs providing physiologic context for heterogeneous and often rapidly evolving patients, MC-BEC presents opportunities to study the uniquely dynamic and complex nature of emergency care.

MC-BEC emphasizes clinically relevant downstream tasks at multiple timescales, specifically predicting patient decompensation (within minutes), disposition (within hours), and ED revisit (within days), and provides a standardized evaluation framework with train-test splits and evaluation metrics. We also provide baselines for each task to enable model comparison and evaluation. With MC-BEC, we hope to encourage researchers and clinicians to develop more effective, generalizable, and accessible foundation models for EHR analysis in Emergency Medicine, ultimately improving patient outcomes and advancing the analysis of real-world EHR data.

## 2 Related Work

### 2.1 Current ED benchmarks are not multimodal

Existing EHR datasets for ED or critically ill patients are often limited to structured EHR data and intermittent vital sign recordings. These datasets fail to capture the comprehensive multimodal information obtained from the intensive evaluation and monitoring of ED patients. To our knowledge, only two ED-specific benchmarks exist. Xie et al. (2022) [2] introduced an ED benchmark using the MIMIC-IV-ED dataset [3]. While this dataset represents the only publicly available general-purpose ED dataset, it contains only tabular EHR data for all patients, with radiology reports for a subset. EHRShot [4] is the other recent ED benchmark, but its underlying dataset includes only coded data such as ICD diagnosis codes, and is not publicly available.

Due to the lack of robust ED benchmarks, we also reviewed existing critical care benchmarks, since patient monitoring practices in the ED and intensive care unit (ICU) are similar. A comparison in Table 1 shows most ICU datasets also focus on structured EHR data and intermittent vital signs, lacking free text or waveforms. HiRID[5] provides high-resolution physiological data but no text/reports. Strikingly, neither ICU nor ED datasets include electrocardiogram (ECG) and photoplethysmograph (PPG) waveforms, which represent essential data on the physiology of critically ill patients.

To address the lack of multimodal ED benchmarks, we propose MC-BEC, using a novel multimodal dataset including vital signs, continuous physiologic waveforms (ECG, PPG, respiration), radiology reports, and diverse structured EHR data. This supports improved evaluation of model performance on a wider range of salient patient information.

### 2.2 Current ED benchmarks are not suitable for generalist medical AI

Existing medical AI benchmarks fall short in comprehensively evaluating generalist medical AI (GMAI) and clinical foundation models. GMAI was recently proposed as a goal for foundation models in healthcare and medicine [6]. GMAI would perform a wide range of medical tasks and flexibly interpret different combinations of data modalities, enabled by foundation models' capability to learn broadly useful data representations from massive pretraining [7]. However, as discussed in a recent survey [8], current benchmarks focus narrowly on accuracy for predefined tasks, inadequate for evaluating key GMAI capabilities. A major limitation is the lack of assessment for multimodal

---

[1]Multimodal Clinical Monitoring in the Emergency Department (MC-MED-v0)

integration (Table 1). While GMAI is designed to integrate diverse data modalities, performance can decline with more modalities [9]. Yet accuracy metrics alone will not expose these nuances. Current benchmarks also rarely evaluate how models handle missing modalities, though incomplete data is pervasive in real-world EHRs.

To address these gaps, benchmarks must move beyond accuracy to rigorously assess multimodal performance, tolerance to missing data, and other facets critical for GMAI. The proposed MC-BEC benchmark exemplifies this comprehensive approach through evaluating multitask learning, multimodal performance, and fairness analyses beyond predictive accuracy alone. Rethinking evaluation is imperative as GMAI diverges from narrow benchmarks of the past toward more expansive and integrative capabilities. Robust benchmarks will be central to steering progress in this promising new direction.

Table 1: Comparison of critical and emergency care EHR benchmarks. MT Learning stands for multitask learning; MM Analysis stands for multimodal analysis; the number of patients and visits of MIMIC-III excludes neonatal patients.

| Benchmark | Eval. Beyond Acc. | | | Source | | | | | Data Modalities | | | |
|---|---|---|---|---|---|---|---|---|---|---|---|---|
| | MT Learning | MM Analysis | Fairness | Dataset | ICU | ED | Num. patients | Num. visits | EHR Codes | Free-Text Notes | Vital Signs | ECG/PPG Waveform |
| MIMIC-Extract[10] | | | | MIMIC-III[11] | x | | 39K$^*$ | 53K$^*$ | x | x | x | |
| Purushotham 2018[12] | | | | | | | | | | | | |
| Harutyunyan 2019[13] | x | | | | | | | | | | | |
| Xie 2022[2] | | | | MIMIC-IV-ED (v1.0)[14] | | x | 217K | 449K | x | x | x | |
| Gupta 2022[15] | | | x | MIMIC-IV (v1.0) [16, 17] | x | x | 257K | 524k | x | x | x | |
| eICU[18] | | | | eICU[19] | x | | 139K | 201K | x | | x | |
| EHR PT[20] | | x | | MIMIC-III[11] / eICU[18] | | | | | | | | |
| HiRID-ICU[21] | | | | HiRID (v1.1.1)[5] | x | | 34K | 56K | x | | x | |
| EHRSHOT[4] | | | | Stanford Med. | x | x | 7k | 894k | x | | | |
| **MC-BEC** | x | x | x | Stanford EM | | x | 63K | 102K | x | x | x | x |

## 2.3  GMAI for multimodal EHR data

Our baseline model is inspired by related work in the field. A common approach to foundation models is to first pretrain them on tasks with abundant data to generate robust representations, then to fine-tune on downstream tasks. ClinicalBERT [22] is a widely used pretrained model specifically designed for clinical text embeddings. Categorial EHR data also has pretraining options like BEHRT[23], CLMBR[24] and MOTOR [25], which predict future EHR codes based on past codes. However, these representations of EHR codes depend on the code formats used during training, limiting transferability and robustness to heterogeneous EHR formats. Instead, we employed CodeEmb's [26] approach that uses pretrained text embeddings associated with EHR code descriptions regardless of the format or code system used by the EHR. For fine-tuning on multimodal data, we adopt the HAIM framework [27], using a modular ML pipeline that integrates modality-specific pretrained embeddings.

## 2.4  Multitask learning

Our multitask training approach draws inspiration from prompting techniques in language models. Rather than using task-specific prediction heads, we employ a unified task representation combined with task-specific queries. This allows creating broadly capable models without architectural changes for new tasks.

Conventional multitask learning often designates separate prediction heads for each task, as in MOTOR which has heads for distinct medical predictions[25]. However, the promising avenue of using unified task representations combined with task-specific queries has gained traction, as showcased in studies like OFS [28] and Perceiver IO [29]. We enhance our baseline LightGBM model [30] by incorporating contextual embeddings from the BERT model. For each task, we concatenate BERT embeddings of the task-specific queries with other input features. This integrates BERT's rich semantic understanding, allowing the model to adeptly differentiate tasks within the shared

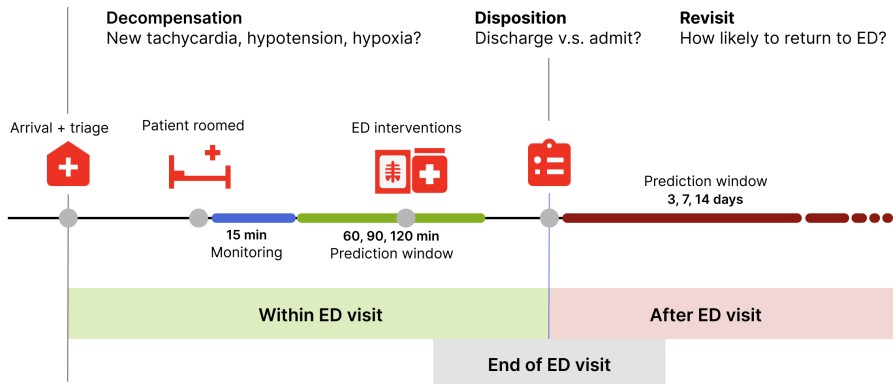

Figure 1: MC-BEC evaluates foundation model performance on predictions of ED patient decompensation, disposition, and revisit, using a unique multimodal dataset of 102,731 monitored ED visits.

embedding space. This provides flexibility to expand to new prediction tasks without architectural changes.

## 3 Benchmark

MC-BEC is a benchmark for evaluating clinical foundation models/GMAI on multimodal EHR data in the ED setting. It provides three clinically relevant prediction tasks spanning different stages of an ED visit: prediction of near-term decompensation early in the visit, prediction of disposition at the end of a visit, and prediction of revisit after the visit. MC-BEC includes evaluation metrics beyond prediction performance, capturing modality interaction, fairness, and consistency of predictions with respect to time and modalities.

### 3.1 Data source

The MC-BEC dataset consists of 102,731 ED visits made by 63,389 patients between September, 2020 and September, 2022. The dataset is IRB approved, with a waiver of informed consent for retrospective research on de-identified data. The de-identified data for each ED visit spans the entire visit from department arrival to departure, including triage after ED arrival, rooming and initiation of monitoring, and all interventions and results up to the point of departure from the ED. Patients in the dataset have multiple ED visits, but MC-BEC ensures no patient overlap (i.e., visits from the same patient) across the training, testing and validation cohorts. Unlike previous ED datasets, MC-BEC contains both categorical and unstructured clinical data. These modalities and data structures are described below.

**Categorical data with single observation:** chief complaint, triage acuity level, gender, race, ethnicity, means of arrival, disposition of most recent visit, disposition of current visit, and payor class.

**Categorical data with repeated observations:** current and previous ICD-10 diagnosis codes with timestamps and corresponding text descriptions; home medications and medications administered during visit with IDs, names, and timestamps of administration; orders placed during visit with IDs, names, and timestamps; lab results with IDs, names, result values, and timestamps.

**Numeric data:** age, hours since previous visit, and vital signs including heart rate, respiratory rate, blood pressure, oxygen saturation, and temperature.

**Time-series data:** continuously recorded vital signs (heart rate, respiratory rate, oxygen saturation) and secondary features (heart rate variability, pulse transit time, perfusion index) throughout the patient's visit. These measurements were averaged over 1-minute intervals.

**Waveform data:** electrocardiogram (ECG), photoplethysmogram (PPG) and respiratory waveforms.

**Free-text data:** radiology reports for imaging studies ordered during the ED visit.

## 3.2 Benchmark tasks

We chose decompensation, disposition, and revisit as prediction tasks for MC-BEC. These tasks have high clinical and operational relevance in the ED setting: early identification of patients at risk of decompensation allows healthcare providers to allocate appropriate resources; accurate disposition prediction helps optimize resource allocation, bed management, and patient flow within the hospital system; and understanding the probability of revisit allows healthcare providers to identify patients who may require closer follow-up or additional interventions to prevent complications or ensure proper continuity of care. These prediction tasks collectively encompass the entire timeframe of an ED visit, focusing on different aspects in the patient's journey through the ED encounter (Figure 1). We provide brief descriptions of each task as defined for our benchmark:

1. **Decompensation**: The goal of this binary classification task is to predict which patients are likely to experience clinical decompensation, defined as new onset of tachycardia (heart rate [HR] > 110), hypotension (mean arterial pressure [MAP] < 65mmHg), or hypoxia (oxygen saturation by pulse oximetry [SpO2] < 90%) in patients with initially normal vital signs. The task uses the first 15 minutes of data acquired after the patient is roomed (the assessment period) to predict decompensation in a 60, 90 or 120 minute-window (evaluation period) following the assessment period. Patients with abnormal vital signs at triage or at any point during the assessment period are excluded from this task, in order to prevent trivial predictions (e.g., the prediction of future hypotension in an already hypotensive patient).

2. **Disposition:** The objective is to predict the binary outcome of a patient's disposition from the ED: whether they will be discharged home or admitted to the hospital. This is a summative clinical decision that reflects the patient's overall clinical stability and risk as determined by the totality of evidence accrued during the visit. To avoid data leakage, any information that could reveal the disposition decision (such as a consult order to an admitting service) is excluded for this task.

3. **Revisit:** The goal is to predict whether a patient will revisit the ED within a 3, 7, or 14-day period after being discharged. ED revisits are a common quality metric, because return visits can sometimes suggest incomplete workup or inappropriate disposition (i.e., a patient sent home who should have been admitted). All data from a visit can be used to make the revisit prediction, and no patients or visits from the dataset need to be excluded for this task.

## 3.3 Evaluation framework

**Prediction performance.** We propose to evaluate prediction performance with area under the precision-recall curve (AUPRC) as a threshold-independent metric. AUPRC is a clinically meaningful metric because it reflects the model's performance in correctly identifying positive instances while minimizing false positives. In medical scenarios, correctly identifying positive cases is crucial, as it ensures accurate diagnosis or prediction of a particular condition. AURPC is also particularly important when dealing with unbalanced data, which is common among medical datasets, including ours.

**Monotonicity in modalities.** We propose to assess the performance change of a model when increasing the data modalities available for training. We hypothesize that a well-designed multimodal model should not perform worse when more data modalities are used for training. However, such monotonicity of performance in modalities is not always observed. For example, [9] found that a unimodal network performed better than the multimodal network obtained through joint training. A reduction in performance caused by the inclusion of more modalities may be attributed to modality competition during training, wherein the model learns to rely on only a subset of modalities to make predictions [31]. The evaluation of monotonicity in modalities can help guide the development of more effective and robust multimodal models.

We evaluate monotonicity in modalities $m_{modalities}$ by calculating the concordance index $C$ between the actual performance $p_m(i)$ and the theoretical ground truth which would expect better performance $y_m(i)$ for increasing numbers $n$ of modalities $m$:

$$y_{m1} \leq y_{m2} \leq \cdots \leq y_{m(n-1)} \leq y_{m(n)} \tag{1}$$

$$m_{modalities} = C = \frac{\sum_{i=1}^{n} \sum_{j=i+1}^{n} f(y_i, y_j, p_i, p_j)}{n(n-1)/2} \tag{2}$$

$$f(y_i, y_j, p_i, p_j) = \begin{cases} 1 & \text{if } (y_i < y_j) \cdot (p_i < p_j) + (y_i > y_j) \cdot (p_i > p_j) \\ 0.5 & p_i = p_j \\ 0 & \text{otherwise} \end{cases} \qquad (3)$$

**Robustness against missing modalities.** We evaluate the model's robustness and generalization capabilities when one or more modalities of data used in training are unavailable in testing. This metric encourages the development of models with more reliable performance in real-world applications where not all training modalities will be present in all cases.

**Bias.** We evaluate bias as a crucial metric to detect systematic errors or prejudices in the model's predictions or outputs. MC-BEC reflects the ED, where false negative prediction of adverse events can result in severe negative health outcomes. Incorrect disposition decisions might exacerbate health issues; misjudged revisit predictions can overlook preventive care needs; and unforeseen clinical decompensation, like hypotension or hypoxia, can be life-threatening. Hence, we chose the true positive rate (TPR) difference as our primary bias metric. Using a sensitivity threshold of 0.85 on the validation set, we assess TPR differences between different demographic groups such as age, gender, race, and ethnicity. Our approach for evaluating model fairness seeks to identify underdiagnosis bias, which is arguably most relevant and consequential in healthcare settings [32].

## 4 Experiments

We developed and evaluated baseline models to demonstrate the use of MC-BEC. Our models used pretrained text and waveform embeddings and employed a modular approach for modality fusion. We trained multimodal representations using LightGBM [30], as gradient boosting ensembles are often highly performant in clinical tasks [33, 34, 35, 36]. We also proposed a novel multitask training schema with LightGBM and compared its performance against models trained on individual tasks. Our aim is to provide an example of MC-BEC evaluation, and a reference for future models evaluated with this benchmark.

### 4.1 Baseline models

**Featurization of multimodal data.** In our baseline model, we employ a modular approach in multimodal fusion that combines different embeddings to create a multimodal feature representation (Figure 2). The data featurization module is highly specialized for a given modality. Singly observed categorical and numeric data are fed directly into the fusion and prediction modules. All other data modalities undergo an additional step of featurization. For continuous vital sign monitoring, we calculate minimum and maximum values and linear trends. We manually engineer features such as heart rate variability, pulse transit time, and perfusion index from the ECG and PPG waveforms, and produce ECG embeddings from a pre-trained transformer [37]. To encode EHR codes with text descriptions such as diagnoses, medications, and lab results, we leverage ClinicalBERT embeddings to capture the richer and more general information provided by the text description, rather than relying on one-hot embeddings for the codes themselves. For orders, we employ Word2Vec-like embeddings, which are trained based on the co-occurrence of order pairs within patient visits. Lastly, from radiology reports, we produce text embedding features using the pre-trained RadBERT transformer as outlined in Yan et al.'s work [38]. These diverse information sources—ClinicalBERT, Word2Vec for orders, and RadBERT for radiology reports—are then concatenated to form a comprehensive multimodal embedding representation.

**Multitask learning.** We propose a novel multitask training schema for our LightGBM models inspired by prompting in language models, whereby a model's predictions are guided by simply providing specific task descriptions. Depending on the training schema, whether multitask or single-task, we further concatenate the multimodal representation with a task description encoded by BERT-Tiny [39, 40] embeddings. The task descriptions used are "predict disposition" for disposition prediction; "predict revisit within [D] days" for revisit prediction, where D is 3, 7 or 14 depending on the revisit horizon; and "predict decompensation within [M] minutes" for decompensation prediction, where M is 60, 90 or 120 depending on the decompensation window of interest. We chose to encode the task as a text embedding because this approach will have the capability to generalize for predictions on a non-discrete set of tasks. Finally, we train a LightGBM model to make predictions based on these enriched multimodal representations.

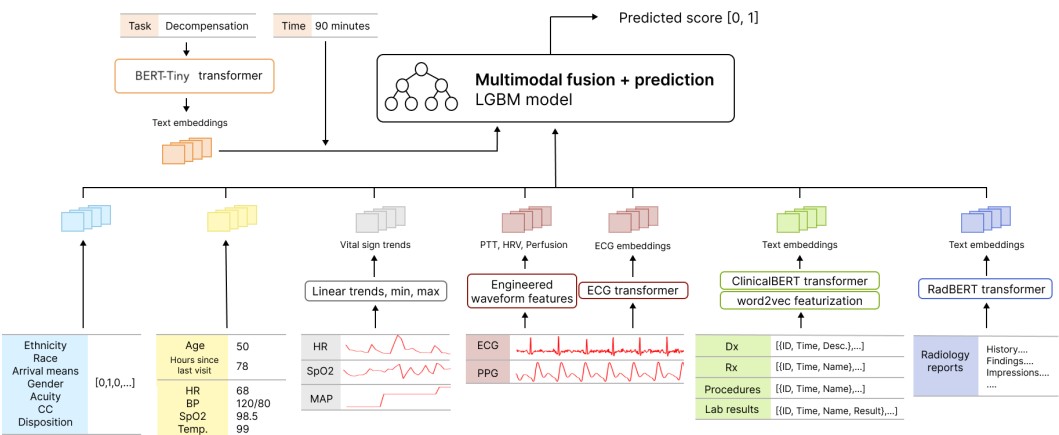

Figure 2: Summary of data modalities represented in MC-BEC and modality-specific featurization strategies.

Table 2: AUPRC of task-specific and multitask models (point estimate and 95% confidence interval).

| Task | Model | Task-specific model | Multitask model | Class Prevalence |
|---|---|---|---|---|
| **Decompensation** | 60 min | 0.33 (0.27 - 0.40) | 0.25 (0.20 - 0.31) | 0.11 |
| | 90 min | 0.35 (0.30 - 0.41) | 0.32 (0.27 - 0.38) | 0.15 |
| | 120 min | 0.40 (0.35 - 0.46) | 0.35 (0.30 - 0.41) | 0.17 |
| **Disposition** | | 0.87 (0.85 - 0.87) | 0.82 (0.81 - 0.83) | 0.37 |
| **Revisit** | 3 days | 0.11 (0.08 - 0.14) | 0.10 (0.08 - 0.13) | 0.05 |
| | 7 days | 0.16 (0.14 - 0.19) | 0.16 (0.14 - 0.19) | 0.08 |
| | 14 days | 0.24 (0.21 - 0.27) | 0.25 (0.22 - 0.28) | 0.12 |

## 4.2 Benchmark results

**Prediction performance.** We compare the performance of models trained on a single task (decompensation, disposition, or revisit) with a model trained simultaneously on all three prediction tasks. Table 2 presents the model performance as assessed by AUPRC using all available data modalities. Overall, task-specific models perform better than the multitask model in decompensation and disposition predictions, but similarly in revisit predictions. For decompensation prediction, the task-specific model has AUPRCs of 0.33, 0.35 and 0.40 (for 60-, 90- and 120-minute prediction windows), while AUPRCs for the multitask model are 0.25, 0.32, and 0.35. Disposition is predicted with AUPRC 0.87 for the task-specific model and 0.82 for the multitask model. When predicting revisit at 3, 7 and 14-day time windows, we report AUPRC scores of 0.11, 0.16, and 0.24 for the task-specific model, and 0.10, 0.16, 0.25 for the multitask prediction model. The prevalence of the positive class in the dataset is provided as baseline AUPRC [41]. Our results indicate slightly worse performance with a multitask prediction model which conforms with the assumption that the single-task prediction models will be better specialized for their specific task. However, the overall performance gap between single and multitask approaches is small. This suggests potential for more advanced multitask modeling to match or surpass the single-task results by better leveraging commonalities and differences between tasks. Recent work has shown promise for sophisticated multitask architectures surpassing single-task models in clinical predictions. For instance, a multitask channel-wise LSTM exceeded single-task models in predicting in-hospital mortality[13]. Future research can use MC-BEC as an avenue to improve upon multitask models.

**Monotonicity in modalities.** To assess monotonicity in modalities, we train the multitask prediction model on an increasing subset of data modalities. We start with a model trained only on categorical and numeric features, then incrementally add features from diagnoses, medications, orders, lab results, radiology reports, continuous monitoring, and waveforms. Table 3 shows the AUPRC of

Table 3: AUPRC of multitask models with incrementally increasing testing and training modalities (point estimate and 95% confidence interval). The modality concordance index measures monotonicity of prediction performance in modalities.

| Model | Decompensation | | | Disposition | Revisit | | |
|---|---|---|---|---|---|---|---|
| | 60 min | 90 min | 120 min | | 3 day | 7 day | 14 day |
| **Modality added** | | | | | | | |
| + Categorical/numerics | 0.23 (0.18 - 0.29) | 0.30 (0.25 - 0.35) | 0.33 (0.28 - 0.39) | 0.71 (0.69 - 0.72) | 0.10 (0.08 - 0.13) | 0.15 (0.13 - 0.18) | 0.25 (0.22 - 0.28) |
| + Diagnoses | 0.23 (0.18 - 0.29) | 0.30 (0.25 - 0.35) | 0.33 (0.28 - 0.39) | 0.70 (0.68 - 0.72) | 0.09 (0.07 - 0.11) | 0.15 (0.13 - 0.18) | 0.24 (0.21 - 0.27) |
| + Medications | 0.22 (0.18 - 0.28) | 0.29 (0.24 - 0.34) | 0.32 (0.28 - 0.38) | 0.71 (0.69 - 0.72) | 0.09 (0.07 - 0.12) | 0.15 (0.13 - 0.18) | 0.24 (0.22 - 0.27) |
| + Orders | **0.18 (0.15 - 0.24)** | **0.24 (0.21 - 0.30)** | **0.27 (0.23 - 0.32)** | 0.79 (0.78 - 0.80) | 0.09 (0.08 - 0.12) | 0.15 (0.13 - 0.17) | 0.24 (0.22 - 0.27) |
| + Labs | **0.16 (0.13 - 0.20)** | **0.22 (0.19 - 0.26)** | **0.25 (0.21 - 0.29)** | 0.79 (0.78 - 0.81) | 0.09 (0.08 - 0.12) | 0.15 (0.13 - 0.18) | 0.24 (0.22 - 0.27) |
| + Radiology | **0.15 (0.13 - 0.19)** | **0.21 (0.18 - 0.25)** | **0.24 (0.21 - 0.29)** | 0.79 (0.78 - 0.80) | 0.10 (0.08 - 0.12) | 0.16 (0.14 - 0.18) | 0.25 (0.22 - 0.28) |
| + Monitoring | 0.24 (0.19 - 0.29) | 0.30 (0.25 - 0.35) | 0.33 (0.28 - 0.39) | 0.80 (0.79 - 0.82) | 0.10 (0.08 - 0.12) | 0.16 (0.14 - 0.18) | 0.25 (0.22 - 0.28) |
| + Waveforms | 0.25 (0.20 - 0.31) | 0.32 (0.27 - 0.38) | 0.35 (0.30 - 0.41) | 0.82 (0.81 - 0.83) | 0.10 (0.08 - 0.13) | 0.16 (0.14 - 0.19) | 0.25 (0.22 - 0.28) |
| **Modality concordance index** | 0.50 | 0.46 | 0.43 | 0.89 | 0.71 | 0.79 | 0.82 |

Table 4: AUPRC for multitask models with one modality missing in test data, demonstrating robustness to missing modalities (point estimate and 95% confidence interval).

| Model | Decompensation | | | Disposition | Revisit | | |
|---|---|---|---|---|---|---|---|
| | 60 min | 90 min | 120 min | | 3 day | 7 day | 14 day |
| **All modalities** | **0.25 (0.20 - 0.31)** | **0.32 (0.27 - 0.38)** | **0.35 (0.30 - 0.41)** | **0.82 (0.81 - 0.83)** | **0.10 (0.08 - 0.13)** | **0.16 (0.14 - 0.19)** | **0.25 (0.22 - 0.28)** |
| **Missing modality** | | | | | | | |
| - Categorical/numerics | 0.26 (0.21 - 0.32) | 0.32 (0.27 - 0.38) | 0.35 (0.30 - 0.41) | 0.81 (0.80 - 0.83) | 0.06 (0.05 - 0.07) | 0.10 (0.09 - 0.11) | 0.17 (0.16 - 0.19) |
| - Diagnoses | 0.25 (0.20 - 0.31) | 0.32 (0.27 - 0.38) | 0.35 (0.30 - 0.41) | 0.82 (0.81 - 0.83) | **0.10 (0.08 - 0.13)** | **0.16 (0.14 - 0.19)** | **0.25 (0.22 - 0.28)** |
| - Medications | 0.25 (0.20 - 0.31) | 0.32 (0.27 - 0.37) | 0.35 (0.30 - 0.40) | 0.82 (0.81 - 0.83) | 0.08 (0.07 - 0.11) | 0.13 (0.11 - 0.15) | 0.23 (0.21 - 0.26) |
| - Orders | 0.26 (0.21 - 0.32) | 0.34 (0.28 - 0.40) | 0.37 (0.32 - 0.43) | **0.73 (0.72 - 0.75)** | 0.07 (0.05 - 0.09) | 0.11 (0.10 - 0.13) | 0.23 (0.21 - 0.25) |
| - Labs | 0.26 (0.21 - 0.33) | 0.33 (0.28 - 0.39) | 0.36 (0.31 - 0.42) | 0.81 (0.80 - 0.82) | 0.08 (0.06 - 0.10) | 0.13 (0.11 - 0.15) | 0.23 (0.21 - 0.25) |
| - Radiology | 0.25 (0.20 - 0.31) | 0.32 (0.27 - 0.37) | 0.35 (0.30 - 0.40) | 0.80 (0.79 - 0.81) | 0.07 (0.06 - 0.10) | 0.12 (0.11 - 0.14) | 0.23 (0.21 - 0.25) |
| - Monitoring | **0.14 (0.12 - 0.17)** | **0.20 (0.18 - 0.24)** | **0.23 (0.20 - 0.27)** | 0.82 (0.81 - 0.83) | 0.08 (0.06 - 0.10) | 0.13 (0.11 - 0.15) | 0.23 (0.21 - 0.25) |
| - Waveforms | 0.24 (0.19 - 0.29) | 0.31 (0.26 - 0.36) | 0.34 (0.29 - 0.39) | 0.82 (0.81 - 0.83) | 0.08 (0.06 - 0.10) | 0.13 (0.11 - 0.15) | 0.23 (0.21 - 0.26) |
| **Max AUPRC difference** | 0.12 | 0.13 | 0.14 | 0.09 | 0.04 | 0.06 | 0.08 |

models trained with increasing numbers of modalities, along with the modality concordance index. Disposition prediction demonstrates strong monotonicity in modalities with a modality concordance index of 0.86. This indicates that disposition prediction benefits from incorporating more data modalities. Revisit predictions demonstrate moderate monotonicity, with concordance indices of 0.71, 0.79, and 0.82 for the 3, 7, and 14-day windows. Decompensation predictions show the worst monotonicity among all tasks, with notably decreased performance when adding orders, labs, and radiology reports data. This poor monotonicity in modalities is reflected in low concordance indices of 0.50, 0.46, and 0.43 for the 60, 90, and 120-minute windows. However, the performance does increase when continuous monitoring and waveforms are added, which aligns with the expectation that continuous monitoring and waveforms of vital signs are predictive of impending decompensation events.

**Robustness against missing modalities.** We assess the robustness of the multitask benchmark model against missing modalities during inference by dropping individual modalities and comparing the difference in AUPRC between using all modalities versus missing modalities (Table 4). Although revisit predictions do not have the largest maximum AUPRC difference, with values of 0.04, 0.06, and 0.08 for the 3, 7, and 30-day windows, they are more likely to see a performance decrease when missing modalities. Specifically, revisit predictions show notable performance drops when all modalities except diagnoses are missing, compared to the substantial drops for decompensation predictions only when continuous monitoring data is missing and for disposition predictions only when orders data is missing. Furthermore, the performance gap widens in the worst case of missing modalities over the longer time windows for both revisit and decompensation predictions, with maximum AUPRC differences of 0.12, 0.13, and 0.14 for the latter. Qualitatively, the impact on model performance when omitting modalities varies significantly depending on the prediction task. Overall, our results suggest that the benchmark predictions are reasonably robust to missing modalities during inference.

**Bias.** Lastly, in table 5, we report mean TPR differences for the multitask prediction model when applied to different demographic groups, specifically based on patient age, gender, race, and ethnicity. The greatest absolute disparities in mean TPR emerged when comparing different racial groups, with a mean TPR difference of 0.11 for 3-day revisit prediction, followed by a mean TPR difference of 0.09 for 7-day revisit prediction. Notable model bias was also observed between gender groups for 90 and 120-minute decompensation predictions, with TPR differences of 0.08 and 0.09. Additionally, there was a sizable TPR difference of 0.07 between age groups for 60-minute decompensation predictions.

Table 5: Evaluation of model bias across demographic groups. True positive rate (TPR) and absolute TPR difference between different groups are reported below.

| Demographic Group | | Decompensation | | | Disposition | Revisit | | |
|---|---|---|---|---|---|---|---|---|
| | | 60 min | 90 min | 120 min | | 3 day | 7 day | 14 day |
| **Age** | < 55 | 0.87 | 0.86 | 0.88 | 0.93 | 0.69 | 0.75 | 0.88 |
| | >= 55 | 0.94 | 0.91 | 0.92 | 0.97 | 0.65 | 0.79 | 0.92 |
| | TPR difference | **0.07** | 0.05 | 0.05 | 0.04 | 0.04 | 0.03 | 0.04 |
| **Gender** | Male | 0.92 | 0.93 | 0.94 | 0.96 | 0.68 | 0.79 | 0.91 |
| | Female | 0.88 | 0.85 | 0.85 | 0.95 | 0.67 | 0.74 | 0.87 |
| | TPR difference | 0.05 | **0.08** | **0.09** | 0.01 | 0.01 | 0.04 | 0.04 |
| **Race** | White | 0.88 | 0.88 | 0.88 | 0.96 | 0.63 | 0.77 | 0.88 |
| | Black/African American | 1.00 | 0.90 | 0.95 | 0.95 | 0.82 | 0.90 | 0.94 |
| | Asian | 0.88 | 0.87 | 0.88 | 0.96 | 0.64 | 0.73 | 0.85 |
| | Native Hawaiian/Other Pacific Islander | 1.00 | 0.86 | 1.00 | 0.93 | 0.80 | 0.86 | 0.81 |
| | Other | 0.90 | 0.90 | 0.91 | 0.95 | 0.70 | 0.74 | 0.90 |
| | TPR difference | 0.07 | 0.02 | 0.06 | 0.01 | **0.11** | **0.09** | 0.06 |
| **Ethnicity** | Non-Hispanic/Non-Latino | 0.89 | 0.88 | 0.89 | 0.96 | 0.68 | 0.77 | 0.88 |
| | Hispanic/Latino | 0.92 | 0.91 | 0.92 | 0.94 | 0.66 | 0.75 | 0.91 |
| | TPR difference | 0.03 | 0.03 | 0.03 | 0.01 | 0.02 | 0.02 | 0.03 |

We also present supplementary fairness metrics like Statistical Parity Difference (SPD), Disparate Impact (DI), and Equal Opportunity Difference (EOD) between minority and majority cohorts in Table 15 in the Appendix. Models applied to the MC-BEC benchmark should be attentive to potential biases as assessed by our framework, and any identified biases must be carefully considered before deployment to ensure fairness and equity.

## 5    Limitations

While the benchmark provides valuable insights, MC-BEC has limitations. The benchmark does not assess the potential of foundation models in automating novel clinical tasks, such as summarizing all data obtained throughout a visit into an easy-to-understand report for the provider or patient. Future iterations of the benchmark should include novel tasks specific to the ED environment to further highlight the capabilities of clinical foundation models. Although the benchmark evaluates bias on measured patient characteristics, it does not account directly for socioeconomic status. Due to privacy and legal concerns, socioeconomic information, such as income or education level, are not included in the underlying dataset. However, the dataset does include visit payor information, which is correlated with socioeconomic status (for instance, patients whose visits are paid by Medicaid are more likely to have lower socioeconomic status). Finally, we used a relatively simple baseline model to demonstrate the benchmark. Although this choice may limit the model's performance, the primary contribution of this work is the proposal of a new benchmark using a novel multimodal dataset. LightGBM, despite its simplicity, remains a commonly used and highly performant approach for many clinical tasks. Therefore, the baseline model serves not only as a demonstration of the benchmark but also as a reasonable reference for future models.

## 6    Conclusion and Future Work

Our work introduces a novel benchmark (MC-BEC), enabling robust multitask evaluation of prediction models for Emergency Medicine, using a unique multimodal clinical dataset. We evaluate multiple baseline models, and compare the performance of multitask and task-specific models. We present an evaluation schema suited to the demands of multitask clinical predictions on complex multimodal data, which assesses a model's ability to make full use of multimodal data, to make robust

predictions despite missing data, and to make fair and unbiased predictions across demographic groups.

The benchmark models introduced in this work employed an explicit data featurization step, which highlights the challenges of working with diverse, complex and data-rich modalities. We expect future models applied to MC-BEC to leverage alternative approaches, such as with deep end-to-end trainable networks, and integration of a temporal dimension both within and across visits. MC-BEC is centered on predictions of decompensation, disposition, and revisit because these encompass the entire time scale of ED visit prediction tasks. However, future work in benchmarks for generalizable ED prediction models should also aim to evaluate prediction over a non-discrete set of tasks, such as prediction of therapeutic complications or need for post-discharge follow-up. This would enable more granular and comprehensive predictions that could help providers to target and prioritize interventions.

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

# Appendix

## 1 Training Details

The dataset was divided into training, testing, and validation sets, as shown in Table 6. During training, we applied sample weights to visits based on the frequency of the label classes. Specifically, the weight for each visit was directly correlated with the frequency of the predominant class and inversely correlated with the frequency of its own class. The distribution of labels is presented in Table 7. It is worth noting that the prevalence of revisits was calculated only among visits with a discharge disposition, since revisits are only relevant for patients discharged from the ED.

Table 6: Distribution of Dataset into Train, Validation, and Test Sets

| Dataset | Number of Samples | Percentage (%) |
|---|---|---|
| Train | 82,438 | 80.25 |
| Validation | 10,235 | 9.96 |
| Test | 10,058 | 9.79 |
| Total | 102,731 | 100 |

Table 7: Label distribution for different tasks

| Category | Task | Class Prevalence |
|---|---|---|
| Disposition | n/a | 0.37 |
| Revisit | 3 days | 0.05 |
| | 7 days | 0.08 |
| | 14 Days | 0.12 |
| Decompensation | 60 mins | 0.11 |
| | 90 mins | 0.15 |
| | 120 mins | 0.17 |

**Hyperparameter Tuning**

We employed the Tree-structured Parzen Estimator (TPE) for hyperparameter tuning, as implemented in the Optuna optimization framework [42]. TPE, a Bayesian optimization approach, leverages outcomes from preceding trials to inform the selection of subsequent hyperparameter values. In this framework, Optuna adeptly pinpoints areas within the hyperparameter space anticipated to enhance the objective function's value. For each model, we executed 20 trials using the hyperparameter spaces delineated in Table 8.

Table 8: Search space used for hyperparameter tuning and the optimal values for the multitask model trained on all data modalities

| Hyperparameters | Search Space | Optimal Values |
|---|---|---|
| **LightGBM** | | |
| lambda_l1 | [1e-8, 10] | 1.84 |
| lambda_l2 | [1e-8, 10] | 0.07 |
| num_leaves | [2, 306] | 134 |
| feature_fraction | [0.4, 1.0] | 0.58 |
| bagging_fraction | [0.4, 1.0] | 0.79 |
| bagging_freq | [1, 7] | 4 |
| max_depth | [-1, 15] | 6 |
| learning_rate | [1e-4, 1] | 0.09 |
| **XGBoost** | | |
| n_estimators | [100, 10,000] | 734 |
| learning_rate | [0.001, 0.5] | 0.04 |
| max_depth | [1, 10] | 7 |
| subsample | [0.25, 0.75] | 0.46 |
| colsample_bytree | [0.05, 0.5] | 0.30 |
| colsample_bylevel | [0.05, 0.5] | 0.31 |
| **Random Forest** | | |
| n_estimators | [100, 10,000] | 9864 |
| max_depth | [1, 10] | 10 |
| min_samples_split | [2, 11] | 2 |
| min_samples_leaf | [1, 10] | 10 |

## 2 Additional Experiments

### 2.1 Random Forest and XGBoost

In addition to LightGBM, we trained Random Forest and XGBoost[43] models for the same clinical prediction tasks, comparing single-task and multitask performance. All modalities were used for training. Results for Random Forest are shown in Table 9, XGBoost in Table 10, and LightGBM again in Table 11 for comparison.

LightGBM outperforms both Random Forest and XGBoost in most tasks. Therefore, we focused the remaining experiments (monotonicity in modalities, robustness to missing modalities, and bias evaluation) on only the LightGBM models.

### 2.2 CLMBR

We conducted additional experiments to explore integrating past records of ED visits for a patient into their EHR representations. We trained a clinical language model-based representation (CLMBR) [24] model on the structured portion of the EHR data. CLMBR treats a patient's longitudinal EHR as a "document" comprising sequences of diagnosis, procedure, medication, and laboratory codes. It learns representations of patient EHRs by predicting future EHR codes based only on past EHR codes. We used the EHR representations learned by CLMBR as input to LightGBM models to predict the clinical tasks (Table 12). As CLMBR is designed for structured EHR data, this experiment could only utilize the structured portion of the dataset, preventing direct comparison to other experiments using all modalities. However, the model's performance is not too far off from others in predicting revisits and disposition. Its performance is lower in predicting decompensation, but that is because it doesn't have access to the continuous monitoring and waveform modalities, which are important in predicting decompensation. Overall, this experiment demonstrates the potential in integrating a patient's longitudinal history into learned EHR representations for improved predictive modeling, an area for future work.

Table 9: AUPRC performance of task-specific and multitask models (point estimate and 95% confidence interval) for Random Forest

| Task \ Model | | Task-specific model | Multitask model | Class Prevalence |
|---|---|---|---|---|
| **Decompensation** | 60 min | 0.21 (0.17 - 0.26) | 0.17 (0.14 - 0.21) | 0.11 |
| | 90 min | 0.28 (0.24 - 0.34) | 0.22 (0.19 - 0.27) | 0.15 |
| | 120 min | 0.31 (0.27 - 0.37) | 0.25 (0.21 - 0.30) | 0.17 |
| **Disposition** | | 0.83 (0.82 - 0.84) | 0.80 (0.79 - 0.81) | 0.37 |
| **Revisit** | 3 days | 0.09 (0.07 - 0.12) | 0.08 (0.07 - 0.11) | 0.05 |
| | 7 days | 0.14 (0.12 - 0.16) | 0.13 (0.12 - 0.16) | 0.08 |
| | 14 days | 0.21 (0.19 - 0.23) | 0.20 (0.18 - 0.23) | 0.12 |

Table 10: AUPRC performance of task-specific and multitask models (point estimate and 95% confidence interval) for XGBoost

| Task | Model | Task-specific model | Multitask model | Class Prevalence |
|---|---|---|---|---|
| **Decompensation** | 60 min | 0.26 (0.21 - 0.33) | 0.23 (0.19 - 0.28) | 0.11 |
| | 90 min | 0.31 (0.26 - 0.37) | 0.30 (0.25 - 0.35) | 0.15 |
| | 120 min | 0.28 (0.24 - 0.34) | 0.33 (0.28 - 0.38) | 0.17 |
| **Disposition** | | 0.88 (0.87 - 0.89) | 0.85 (0.84 - 0.86) | 0.37 |
| **Revisit** | 3 days | 0.08 (0.06 - 0.11) | 0.08 (0.06 - 0.10) | 0.05 |
| | 7 days | 0.14 (0.12 - 0.16) | 0.14 (0.12 - 0.16) | 0.08 |
| | 14 days | 0.21 (0.19 - 0.24) | 0.22 (0.19 - 0.24) | 0.12 |

Table 11: AUPRC performance of task-specific and multitask models (point estimate and 95% confidence interval) for LightGBM

| Task | Model | Task-specific model | Multitask model | Class Prevalence |
|---|---|---|---|---|
| **Decompensation** | 60 min | 0.33 (0.27 - 0.40) | 0.25 (0.20 - 0.31) | 0.11 |
| | 90 min | 0.35 (0.30 - 0.41) | 0.32 (0.27 - 0.38) | 0.15 |
| | 120 min | 0.40 (0.35 - 0.46) | 0.35 (0.30 - 0.41) | 0.17 |
| **Disposition** | | 0.87 (0.85 - 0.87) | 0.82 (0.81 - 0.83) | 0.37 |
| **Revisit** | 3 days | 0.11 (0.08 - 0.14) | 0.10 (0.08 - 0.13) | 0.05 |
| | 7 days | 0.16 (0.14 - 0.19) | 0.16 (0.14 - 0.19) | 0.08 |
| | 14 days | 0.24 (0.21 - 0.27) | 0.25 (0.22 - 0.28) | 0.12 |

Table 12: AUPRC performance for LightGBM with CLMBR embeddings

| Task | Model | CLMBR + LightGBM | Class Prevalence |
|---|---|---|---|
| **Decompensation** | 60 min | 0.19 (0.15 - 0.24) | 0.11 |
| | 90 min | 0.26 (0.22 - 0.32) | 0.15 |
| | 120 min | 0.28 (0.24 - 0.33) | 0.17 |
| **Disposition** | | 0.81 (0.80 - 0.82) | 0.37 |
| **Revisit** | 3 days | 0.08 (0.07 - 0.10) | 0.05 |
| | 7 days | 0.13 (0.11 - 0.15) | 0.08 |
| | 14 days | 0.22 (0.19 - 0.24) | 0.12 |

# 3 Dataset Details

The distribution of demographics within the dataset can be found in table 13. For a glimpse at the primary data modalities, refer to a sample datapoint presented in table 14.

Table 13: Distribution of Demographic Groups with Counts and Relative Percentages

| Demographic | Number of Visits | Percentage of Total Visits |
|---|---|---|
| **Age** | | |
| 18-30 | 16,159 | 15.73% |
| 30-40 | 15,684 | 15.27% |
| 40-50 | 13,501 | 13.14% |
| 50-60 | 15,355 | 14.95% |
| 60-70 | 15,815 | 15.39% |
| 70-80 | 13,272 | 12.92% |
| 80-90 | 12,361 | 12.03% |
| Unknown | 584 | 0.57% |
| Total | 102,731 | 100.00% |
| **Gender** | | |
| Female | 55,929 | 54.44% |
| Male | 46,774 | 45.53% |
| Unknown | 28 | 0.03% |
| Total | 102,731 | 100.00% |
| **Race** | | |
| American Indian or Alaska Native | 276 | 0.27% |
| Asian | 16,971 | 16.52% |
| Black or African American | 6,590 | 6.41% |
| Declines to State | 475 | 0.46% |
| Native Hawaiian or Other Pacific Islander | 2,147 | 2.09% |
| Other | 34,536 | 33.62% |
| Unknown | 444 | 0.43% |
| White | 41,292 | 40.19% |
| Total | 102,731 | 100.00% |
| **Ethnicity** | | |
| Declines to State | 557 | 0.54% |
| Hispanic/Latino | 28,181 | 27.43% |
| Non-Hispanic/Non-Latino | 73,404 | 71.45% |
| Unknown | 589 | 0.57% |
| Total | 102,731 | 100.00% |

Table 14: Sample of Visits Data including Triage Details, Medications, Prior Diagnosis, Lab Tests, Orders and Radiology Report

| Category | Field | Value |
|---|---|---|
| Visits | Age | 52 |
| | Triage_HR | 95.0 |
| | Triage_RR | 12.0 |
| | Triage_SpO2 | 99.0 |
| | Triage_Temp | 36.4 |
| | Triage_SBP | 142.0 |
| | Triage_DBP | 81.0 |
| | Gender | F |
| | Triage_acuity | 3-Urgent |
| | CC Means_of_arrival_cat | Self |
| | Race | Asian |
| | Ethnicity | Non-Hispanic/Non-Latino |
| Meds | Med_class | DIETARY SUPPLEMENT, MISCELLANEOUS |
| | Med_subclass | Alternative Therapy - Unclassified |
| Dx (Diagnosis) | DescCCS | HIV infection |
| Orders | Procedure | CBC W/O DIFF |
| | Med | MAGNESIUM SULFATE IN WATER 2 GRAM/50 ML (4 %) IV PGBK |
| Labs | Component_name | CALCIUM |
| | Component_abnormal_mod | Normal |
| Rads | Impression | The appendix is inflamed, but is favored to be a reactive process in close proximity to the dominant cecal/terminal iletis. Findings were discussed by ___ over the phone with ___ on ___ 19:27. |

Table 15: Evaluation of model fairness across demographic groups. Three metrics - SPD, DI, and EOD averaged across the 7 experiments have been shown. In the ideal equitable scenario, we expect and aim for DI=1 and SPD, EOD = 0 (Adults are defined as individuals between 18-65 years of age. Individuals older than 65 have been categorized as Seniors)

| Minority | Majority | SPD | DI | EOD |
|---|---|---|---|---|
| Seniors | Adults | 0.09 | 1.14 | 0.04 |
| Hispanic/Latino | Non-Hispanic/Non-Latino | 0.00 | 1.01 | 0.01 |
| Other | White | -0.02 | 0.98 | 0.02 |
| Asian | White | -0.02 | 0.96 | -0.01 |
| Asian | Other | 0.00 | 1.00 | -0.03 |
| Black or African American | White | 0.09 | 1.15 | 0.08 |
| Black or African American | Other | 0.11 | 1.18 | 0.07 |
| Black or African American | Asian | 0.12 | 1.20 | 0.09 |
| Native Hawaiian or Other Pacific Islander | White | -0.02 | 0.97 | 0.05 |
| Native Hawaiian or Other Pacific Islander | Other | -0.01 | 0.99 | 0.04 |
| Native Hawaiian or Other Pacific Islander | Asian | 0.00 | 1.01 | 0.06 |
| Native Hawaiian or Other Pacific Islander | Black or African American | -0.12 | 0.84 | -0.03 |
| Male | Female | 0.03 | 1.04 | 0.05 |

