# OpenReview forum: "Multimodal Clinical Benchmark for Emergency Care (MC-BEC): A Comprehensive Benchmark for Evaluating Foundation Models in Emergency Medicine"
_NeurIPS.cc/2023/Track/Datasets_and_Benchmarks — NeurIPS 2023 Datasets and Benchmarks Poster_

### Official Review · Reviewer_PuQz · 2023-07-07
**Comprehensive Emergency Room Benchmark Dataset**

**Rating:** 8
**Confidence:** 4
**Clarity:** yes

**Strengths:**

The size of the dataset is a strength, as well as the comprehensive, multi-modal nature of the data (triage information, prior diagnoses and medications, continuously measured vital signs, electrocardiogram and photoplethysmograph waveforms, orders placed and medications administered throughout the visit, free-text reports of imaging studies, and information on ED diagnosis, disposition, and subsequent revisits.)


Good definitions of the three predicted metrics (decompensation, disposition, and revisit) are provided.

**Additional Feedback:**

none

**Correctness:**

The authors use AUPRC as the primary evaluation metric, arguing that the class imbalance in the dataset (as provided in Table 7) makes this metric more appropriate than AUC.  We respectfully disagree.

While we are in agreement that AUPRC provides useful information, it is a prevalence dependent metric.  Therefore, we encourage the authors to either provide prevalence information for all predictions or to also calculate AUC.  Without this additional information, it is not possible to fully interpret the results provided by the authors.  As an example, without prevalence information it is impossible to determine whether the incidence rate of revisits is changing over the course of the 3, 7 and 14 day periods or the ability of the model to correctly predict the revisit rate is improving.

We accept as valid the author's argument that sensitivity and specificity metric necessitate an operating point which is not being suggested.

**Documentation:**

Updated review:  While the database is not yet available, the authors have clarified that it will be available within the next year.

original thoughts:  Link to dataset is not available.

**Ethics:**

no ethical concerns identified

**Limitations:**

Updated review:  While the dataset offers the opportunity to investigate multiple timescale, the prediction tasks are limited to patient decompensation (within minutes), disposition (within hours), and ED revisit (within days).

in my original review I raised a concern that the dataset was collected over two years in which COVID prevalence in emergency rooms was quite high (2020-2022), and thus, it is unclear whether it will be a useful benchmark for non-COVID emergency care.  Interactively the authors have clarified that that ~6.3% of cases were COVID-related.

Confidence intervals for model performance are given, but are so large as to call into question the model's performance.

**Opportunities For Improvement:**

The authors evaluate prediction performance with Area Under the Precision-Recall curve (AUPRC).  Without information on prevalence, it is impossible to determine whether the model has any utility and to compare the prediction performance across the three metrics.

As it appears that the authors have full information regarding decompensation, disposition, and revisits for all the patients included in the dataset, precision-recall may not be the best evaluation metric; we encourage the authors to evaluate sensitivity and specificity and the area under the receiver-operating curve instead.  However, the reviewer acknowledges that choice of evaluation metric does not negate the utility of the dataset, for which high marks are given.

**Relation To Prior Work:**

Expands an existing ER prediction dataset MIMIC-IV-ED from tabular HER data to include additional multimodal information

**Summary And Contributions:**

The authors propose a Multimodal Clinical Benchmark for Emergency Care (MC-BEC), a dataset composed of 121,978 continuously monitored Emergency Department visits from 72,892 patients between 2020-2022.

---

> ### Author Response · Authors · 2023-08-19
> **Author's Response to Reviewer's Comments and Feedback**
>
> We thank the reviewer for the positive feedback. Here is our point-by-point response:
> - Evaluation Metrics - AUPRC vs AUC-ROC and Sensitivity/Specificity:
>     - Rationale for AUPRC:
>         - Class Imbalance: As highlighted in table 7 of our revised appendix, our dataset faces a pronounced class imbalance. AUPRC, by design, captures performance nuances on the positive class, making it more insightful than AUC-ROC, especially when true positives are of paramount importance.
>         - Clinical Context: Given the significant consequences of false negatives in emergency healthcare, AUPRC, which evaluates precision at various levels of recall, aligns closely with the clinical need to accurately identify positive cases.
>     - Limitations of AUC-ROC: Its sensitivity to class distribution makes it less robust for imbalanced datasets, potentially offering an overly optimistic view.
>     - Limitations of Sensitivity/Specificity: While informative, these metrics necessitate a threshold or operating point. This choice varies based on the clinical scenario, limiting generalizability. However, given the comprehensiveness of our dataset, researchers can derive these metrics to cater to their specific clinical use case.
> - Relevance of Dataset vis-à-vis COVID-19:
>     - A mere 6.3% of visits in our dataset are associated with COVID. Although these can be excluded for generic emergency care research, the unique conditions of the pandemic period also render this dataset valuable for specialized inquiries.
> - Confidence Intervals:
>     - For clarity and transparency, we've now included 95% confidence intervals for our findings.
> - Dataset Documentation:
>     - The dataset link will be made accessible in compliance with NeurIPS datasets and benchmarks timelines.

---

### Official Review · Reviewer_RbWs · 2023-07-20
**Multimodal Clinical Benchmark for Emergency Care (MC-BEC): A Comprehensive Benchmark for Evaluating Predictive Models in Emergency Medicine**

**Rating:** 6
**Confidence:** 5
**Correctness:** Please see Limitations.

**Strengths:**

+ The Multimodal Clinical Benchmark for Emergency Care (MC-BEC) is introduced as a comprehensive evaluation framework for foundation models in Emergency Medicine. With a dataset comprising 72,982 patients and their multiple visits, MC-BEC offers a substantial amount of data for analysis.
+ Each data entry in MC-BEC includes various field information such as prior diagnoses, medications, and lab results. This incorporation of multiple modalities enriches the dataset, further enhancing its comprehensiveness and breadth.
+ In addition to its rich data collection, MC-BEC provides results of downstream tasks encompassing multiple timescales. This feature facilitates the training of models in predicting and analyzing real-world Electronic Health Records (EHR) data, enabling more accurate insights into clinical scenarios.

**Additional Feedback:**

N/A

**Clarity:**

+ Could you please provide a more detailed explanation of LightGBM? It is the model employed in handling multi-modal representations in this paper. Additionally, you mentioned that it differs from previous approaches due to its specific representation head, but further clarification on this aspect would be appreciated.
+ Could you kindly share one or two examples related to the data within the entire dataset? This would greatly assist in conveying the high quality of the data to readers and enhance their understanding.
+ It would be beneficial to conduct additional experiments using different backbones. Merely selecting one approach (LightGBM) might not provide sufficient conviction. By exploring various backbones, we can strengthen the validity and robustness of our findings.

**Documentation:**

No. Lack of data information.

**Ethics:**

No.

**Limitations:**

+ It appears that the dataset used for the study lacks demographic information, including factors such as the percentage of different ethnicities, age groups, and genders. This omission of crucial demographic variables could potentially introduce bias in the observations, particularly when attempting to make generalizations from the overall dataset.
+ In the experimental section, important details are missing, such as information regarding the learning rate, weight decay, and number of epochs used. It is possible that the subpar results obtained by the baseline models could be attributed to inadequate hyperparameter tuning. It would be beneficial for the authors to provide further elaboration on this aspect to enhance the understanding of the experimental setup.

**Opportunities For Improvement:**

- The dataset includes a field labeled 'Revisit,' but this information may not be entirely accurate, as individuals may fail to visit due to reasons other than discomfort. A more effective approach would be to incorporate information regarding the time period leading up to their discomfort.
- There is a minor issue with citation: In section 2.4, LightGBM should be cited as it is the work of others.
- There is a minor spelling mistake: On line 238, 'decompesnation' should be corrected to 'decompensation.'
- There is a minor error in the results: On line 238, the task-specific model should have an AUPRC of 0.29, 0.35, and 0.35, as indicated in table 1.

**Relation To Prior Work:**

Lack of discussion. Please see limitations.

**Summary And Contributions:**

This paper introduces a comprehensive benchmark called the Multimodal Clinical Benchmark for Emergency Care (MC-BEC), designed to evaluate foundation models using a wealth of monitored Emergency Department visits. The benchmark encompasses multimodal data that includes detailed information, serving as a baseline for training multimodal, multi-task models. The proposed dataset aims to foster the development of more advanced foundational models in emergency care.

---

> ### Author Response · Authors · 2023-08-17
> **Addressing the Reviewer's Feedback on Limitations, Clarity, and Prospects for Enhancement in MC-BEC**
>
> Thank you for your thoughtful comments and suggestions. We have addressed them as detailed below.
>
> ### Response to Opportunities For Improvement
> - The "Revisit" label indicates that patients may return to the emergency department for various reasons. Hospital revisits at different time intervals are common quality measures because they can sometimes reflect inadequate evaluation or inappropriate discharge planning. While we would prefer having details about patients' experiences between visits, this information is unavailable. The benchmark can help by enabling the calculation of time between visits and documenting the outcome (disposition) of the return visit to provide more context around these events.
> - Thank you for catching the missing citation for LightGBM, which we have added.
> - Thank you for catching this typo (“decompesnation”), which we have corrected.
> - Thank you for catching the reporting error for the task-specific model in decompensation prediction. We have updated it with the new results after hyperparameter tuning. We also include 95% confidence intervals for all such results.
>
>
> ### Response to Limitations
> - We thank the reviewer for the suggestion of providing demographic information on the dataset. We have added tables of the age, race, ethnicity, and gender distributions in Appendix 2.
> - We appreciate the suggestion of providing hyperparameter tuning details. We have expanded our hyperparameter tuning strategy using a Bayesian optimization approach, implemented with the open-source framework called Optuna. Details on the search space for the hyperparameters are now provided in Appendix 1.
>
> ### Response to Clarity
> - The reviewer asks us to “provide a more detailed explanation of LightGBM”.
>     - Yes, the model is employed in handling multimodal representations as follows (this has been discussed in the updated section 4.1 of the paper)
>     - The input for our LightGBM model involves combining embeddings from various modalities, such as BERT-based text embeddings for radiology reports, descriptions of diagnoses, medications, and labs, transformer-based ECG embeddings, word2vec embeddings for orders trained on the co-occurrence of order pairs in visits, one-hot encoding for categorical EHR data, along with features derived from continuous vital signs.
>     - Task description embedding To train all three clinical binary prediction tasks simultaneously, while being able to specify the desired task during inference, we also concatenated task descriptions (for example, “3 days revisit”) encoded by ClinicalBERT to the input above.
>     - During training, the LightGBM model was trained on a mixture of <[Multimodal features | Task description embedding], task-specific label> pairs for all tasks. During inference, we concatenated the corresponding task embedding to the multimodal features to get the prediction for the desired task. Using multimodal representation with task-description embedding for multitask learning with LightGBM is what sets us apart from previous approaches.
> - We appreciate the suggestion of sharing one or two samples. We have added sample data in Appendix 2.1.
> - Following your suggestion to experiment with various backbones, we've detailed our updated results in appendix 1. We explored other tree ensemble models: XGBoost and Random Forest, targeting both individual and multitask applications. After a comprehensive hyperparameter optimization for these models, LightGBM emerged as the superior performer across all tasks. Given this dominance, we chose to proceed with further evaluations, such as assessing monotonicity in modalities, ensuring robustness against absent modalities, and examining bias, exclusively using the LightGBM model.
>
> We have thoroughly considered the reviewer's insightful feedback and implemented revisions to enhance the manuscript as suggested. We are confident that these modifications bolster the quality of the work, and we kindly request your consideration of adjusting the overall score in light of our efforts to address the raised concerns.

---

> > ### Comment · Reviewer_RbWs · 2023-08-19
> > **Response to author rebuttal**
> >
> > Having reviewed the authors' rebuttal, I recognize that they have addressed the majority of my initial concerns. I also commend the authors for their constructive engagement in the discussions with other reviewers. In addition, some of the ideas here may be familiar to readers of the following papers [1, 2, 3]. If these studies have any relevance to the topic at hand, it would be great if the authors would highlight it. Given these considerations, I am inclined to adjust my rating from 5 to 6.
> >
> > [1] You, C., Zhao, R., Liu, F., Dong, S., Chinchali, S., Topcu, U., ... & Duncan, J. (2022). Class-aware adversarial transformers for medical image segmentation. Advances in Neural Information Processing Systems, 35, 29582-29596.
> >
> > [2] Liu, F., Yang, B., You, C., Wu, X., Ge, S., Liu, Z., ... & Clifton, D. (2022). Retrieve, reason, and refine: Generating accurate and faithful patient instructions. Advances in Neural Information Processing Systems, 35, 18864-18877.
> >
> > [3] Liu, F., You, C., Wu, X., Ge, S., & Sun, X. (2021). Auto-encoding knowledge graph for unsupervised medical report generation. Advances in Neural Information Processing Systems, 34, 16266-16279.

---

### Official Review · Reviewer_2TbH · 2023-07-21
**A Large-scale Multimodal Healthcare Dataset**

**Rating:** 6
**Confidence:** 4

**Strengths:**

- A large-scale multi-modal dataset consisting of time-series, text, waveform, .. data. Compared to MIMIC-IV, this dataset has more modalities (more waveform data).
- "current ED benchmarks are not suitable for generalist medical AI" The benchmark proposed more evaluation metrics beyond prediction performance, considered the multi-modality characteristics and the fairness issue.
- Comprehensive benchmarks on all proposed metrics and at timescales from minutes to days. (Table 1,2,3,4)
- Inspired by the development of NLP field, the prompting strategy for multitask learning is somehow novel for healthcare data modeling.

**Additional Feedback:**

Is it a mistake not to upload the supplementary material?

**Clarity:**

The paper is essentially well written and structured but more details should be included in the submission.

**Correctness:**

The paper lack details of how they construct the dataset in addition to the modality constitution. The evaluation methods is appropriate, but the experiment design details are unavailable currently.

**Documentation:**

No. there is no sufficent detail.

**Ethics:**

The authors have ensured de-identification of data. They have discussed the potential for bias in AI systems and have included an evaluation of model fairness in their work. No major ethical concerns are identified.

**Limitations:**

The authors have adequately addressed the limitations of their work and potential negative societal impact. They acknowledge the limitations of the current benchmark including the socioeconomic status bias issue and relatively simple LightGBM model used in the pipeline.

**Opportunities For Improvement:**

- Lack reporting the error bar of the performance (mean and std score)
- The motivation to introduce the metrics: Monotonicity in modalities against missing modalities may not vital. It is also a prediction performance measure to some extent.
- As the main contribution is the proposed dataset itself. The paper lack essential data analysis, including feature (demographics and lab tests) distribution, missing rate and outcome-related labels stats.
- The model used is a bit of weired. Concating the pretrained language model, engineered wavelet features, deep-learning based transformer feature extractors as multimodal features, and use a LightGBM model for binary outcome prediction. It is not clear how to train the ECG transformer, RadBERT transformer, since these models require gradient backwards. (Fig 2)
- In checklist, it is said the training details (e.g., data splits, hyperparameters, how they were chosen) would be included in supplemental material. However, authors dis not submit the supplemental material. The benchmark code, resources, and materials for reproducibility are all unavailable.
- Other fairness measures may also be discussed in addition to the TPR difference which is not one of the most popular fairness measures. (e.g. SPD, DI, EOD, ...)
- The prompting strategy is not clearly clarified other than the pipeline figure in Figure 2 and related work. It also has some limitations since it does not work for various types of prediction tasks (Length-of-stay prediction and mortality outcome prediction and multi-class classification tasks may not share the same prediction head)
- Typos: Line 285: has has
- Line 124, it is more appropriate to cite other ED benchmarks.

**Relation To Prior Work:**

Yes. But the more comparison with other existing ED or datasets may be helpful to mention. (stats like patient and visit counts, label distributions, ...)

**Summary And Contributions:**

The submission presents a multimodal clinical benchmark, the Multimodal Clinical Benchmark for Emergency Care (MC-BEC), designed to evaluate predictive models in emergency medicine.

**[Dataset Constitution]** The benchmark is based on a large-scale dataset of 121,978 continuously monitored Emergency Department visits from 2020-2022 with modalities including demographics, EHR time-series data, vital signs, waveform data (ECG, PPG), textual clinical notes (radiology reports), triage information, etc.

**[Prediction Tasks]** The benchmark focuses on clinically relevant prediction tasks at different timescales (from hours to several days) including predicting patient decompensation, disposition, and ED revisit.

**[Evaluation Metrics]** 1. All three tasks are binary prediction tasks, authors compare prediction performance (AUROC, AUPRC) in the first place; 2. Monotonicity in modalities (with more modalities as model's inputs, better performance should be observed); 3. Robustness against missing modalities; 4. Model fairness performance on sensitive demographic groups.

**[Main Contributions]**: Large-scale 72K patients and 121K visits. (Compared to MIMIC-III: ~35K, MIMIC-IV: ~257K); New metrics with multi-modality, missing modality and fairness issues considered.

---

> ### Author Response · Authors · 2023-08-17
>
> We thank the reviewer for the insightful comments. We address “Opportunities For Improvement” as follows:
>
>   - Regarding the lack of error bars, we have added 95% confidence intervals in _Table 2: AUPRC of task-specific and multi-task models, Table 3: AUPRC of multi-task models with incrementally increasing testing and training modalities, Table 4: AUPRC of multi-task models with one modality missing in test data._
>
> - We appreciate this critical feedback on the lack of data analysis. We have now added dataset statistics in the appendix, including label distribution for different tasks in appendix 1 and demographics distribution in appendix 3.
>
>
> - Regarding the modeling approach and how to train the embedding, we froze the embeddings passed onto the LGBM model, so there was no gradient through which to update the models generating the embeddings. This is a common approach for multimodal medical AI models, leveraging pre-trained feature extractors as part of unified processing for downstream modeling [1].
>
>   - [1] Soenksen, Luis R., et al. "Integrated multimodal artificial intelligence framework for healthcare applications." NPJ digital medicine 5.1 (2022): 149.
>
>
> - We appreciate the reviewer's noticing that our supplemental material was not successfully updated. We have now uploaded the appendix with details on the training procedure, hyperparameter tuning (appendix 1), experiment results with different backbones (appendix 2), and dataset statistics (appendix 3). Additionally, we have included sample data of a patient in appendix 3 for reference.
>
>
> - On the point of “Other fairness measures may also be discussed in addition to the TPR difference,” we have now updated the manuscript with other fairness measures including statistical parity difference (SPD), disparate impact (DI), and equal opportunity difference (EOD) results.
>
>
> - To clarify the prompting strategy, we have updated the Multitask learning paragraph in 4.1 and added the specific prompts used by the model. We append a BERT embedding of the task description to other embeddings as part of the input to the LightGBM model. We acknowledge the limitation the reviewer mentioned concerning the applicability of our strategy to non-binary prediction tasks. We agree that this is an important direction for future work, and believe our benchmark will facilitate progress in modeling approaches to accommodate different types of prediction tasks.
>
>
> - Thank you for catching the typo “has has”, which we have corrected.
>
>
> - We appreciate the advice on citing other ED benchmarks. We have made a comparison to other existing ED benchmarks or datasets in Related Work 2.1 and 2.2. Additionally, we have added a table of EHR benchmarks comparison (Table 1). Note that most of these benchmarks are based on ICU data. To our knowledge, Xie et al. [1] and EHRSHOT [2] are the only ED benchmark that currently exists.
>
>   - [1] Xie, Feng, et al. "Benchmarking emergency department prediction models with machine learning and public electronic health records." Scientific Data 9.1 (2022): 658.
>   - [2] M. Wornow, R. Thapa, E. Steinberg, J. Fries, and N. Shah, “Ehrshot: An ehr benchmark for few-shot evaluation of foundation models,” arXiv preprint arXiv:2307.02028, 2023.

---

> > ### Author Response · Authors · 2023-08-17
> >
> >
> > - We appreciate the feedback on the motivation to introduce the metrics of “Monotonicity in modalities” against “missing modalities” and would like to provide further clarity on the motivation behind our evaluation framework. To evaluate "monotonicity in modalities," we incrementally added features during the training phase. This is in contrast to "robustness against missing modalities" where the model is trained using all modalities but is tested with certain modalities ablated, simulating deployment in heterogeneous data environments. These two metrics were designed to answer two orthogonal research questions.
> >
> >   - The question underpinning monotonicity in modalities is whether adding more information to a model during its training could hurt its performance. Intuitively, one might presume that more information would always equate to better performance. However, empirical evidence suggests otherwise. For instance, Sundrani et al.’s work on predicting patient decompensation in the emergency department shows worse performance when some modalities were included, particularly in predicting the onset of hypotension. The best-performing models in this work were not those trained on all available modalities [1]. Similarly, Wang et al. observed that unimodal networks could sometimes outperform their multi-modal counterparts, possibly due to different modalities being prone to overfitting at varying rates [2]. Huang et al. delve deeper into this phenomenon, proposing a theoretical explanation based on the concept of "modality competition" in multi-modal late-fusion networks [3].
> >
> >   - On the other hand, the "Robustness against missing modalities" metric was developed to address the common real-world scenario where certain modalities present during training might be unavailable at inference. We hypothesize that a good model should be robust against missing modalities during inference, a common scenario in real-world electronic health records (EHRs), where a given visit will feature only a small minority of the tests and medications present in the training set, and may be monitored with only a subset of possible monitoring modalities.
> >
> >   - We hope this provides a clearer perspective on our rationale for introducing these metrics, highlighting their relevance and importance in the broader context of our research.
> >
> >
> >     - [1] Sundrani, Sameer, et al. "Predicting patient decompensation from continuous physiologic monitoring in the emergency department." NPJ Digital Medicine 6.1 (2023): 60.
> >
> >     - [2] Wang, Weiyao, Du Tran, and Matt Feiszli. "What makes training multi-modal classification networks hard?." Proceedings of the IEEE/CVF conference on computer vision and pattern recognition. 2020.
> >
> >     - [3] Huang, Yu, et al. "Modality competition: What makes joint training of multi-modal network fail in deep learning?(provably)." International Conference on Machine Learning. PMLR, 2022.

---

> > > ### Author Response · Authors · 2023-08-17
> > >
> > > We have carefully addressed all of the reviewer's thoughtful comments and made revisions to improve the manuscript accordingly. We believe these changes strengthen the work and
> > > hope the reviewer will consider raising the overall score during this revision to reflect our resolution of the concerns raised.

---

> > > > ### Comment · Reviewer_2TbH · 2023-08-29
> > > > **Majority of comments addressed**
> > > >
> > > > The authors have addressed the majority of my comments in their revision. I have moved my score up in good faith.

---

### Official Review · Reviewer_XBLz · 2023-07-22
**Multimodal Clinical Benchmark for Emergency Care (MC-BEC)**

**Rating:** 7
**Confidence:** 5
**Correctness:** Yes.
**Clarity:** Yes.

**Strengths:**

The first useful Multimodal Clinical Benchmark for Emergency Care.

**Additional Feedback:**

No.

**Documentation:**

The authors can only release the data set one year later since the submission.

**Ethics:**

Yes.

**Limitations:**

The authors do a good job of putting up a benchmark. But the authors fail to convince the reviewer that they provide a good benchmark.

**Opportunities For Improvement:**

(1) As a benchmark paper, the authors mainly focused on data collection. The reviewer like to see the impact of different designs and implementations (or even different data selections) of benchmarks. e.g., data distribution. For example, will the evaluation results differ significantly if the other team of authors can access the same or a different data source?

(2) the authors claimed that Current ED benchmarks are unsuitable for generalist medical AI. The author fails to do this job. The reviewer just wonders why current ED benchmarks are not suitable for generalist medical AI.

**Relation To Prior Work:**

Yes.

**Summary And Contributions:**

(1) the authors propose the Multimodal Clinical Benchmark for Emergency Care (MC-BEC), a comprehensive benchmark for evaluating foundation models in Emergency Medicine using a dataset of 121,978 continuously monitored Emergency Department visits from 2020-2022.

(2) The authors provide performance baselines for each prediction task to enable the evaluation of multimodal, multi-task models.

---

> ### Author Response · Authors · 2023-08-18
>
> We thank the reviewer for the positive feedback. We will respond to the comments in the following:
>
>
> - We thank the reviewer for the thoughtful comment on evaluating the impact of different designs and implementations. To better evaluate the impact of different modeling choices, we have conducted additional experiments using Random Forest and XGBoost backbones. The results are now included in Appendix 2. We observed some performance differences across backbones, though LighGBM still achieved the best overall performance, and the trends between single-task and multitask models were consistent across backbones (i.e. single-task outperform multitasks). These new experiments provide additional insight into how backbone selection influences benchmark performance. We agree that studying how factors like data distribution, model architecture, and training procedures impact benchmark performance is an exciting area for future research. Our current work lays the foundation for such studies by providing this new suite of benchmarks and analysis. We look forward to future work further examining how design and implementation choices influence benchmark results.
>
>
> - Regarding why our benchmark is better suited for generalist medical AI than current benchmarks, we have expanded Sections 2.1 and 2.2 and added Table 1 comparing our benchmark to prior work. There are currently only two ED benchmarks, so we also compare our benchmark to existing ICU benchmarks given both settings involve critically ill patients. Our benchmark advances prior ED/ICU benchmarks in two key aspects - the dataset and the evaluation methods.
>    - First, our dataset is more comprehensive and representative than existing benchmarks. Notably, we are the first to include raw ECG and PPG waveforms. We also include radiology reports, and high-resolution, continuous physiological recordings, which are not always available in other benchmarks. This gives our dataset closer parity to real clinical data that ED physicians utilize compared to other benchmarks.
>    - Second, prior benchmarks focus narrowly on accuracy metrics, which we argue is inadequate for evaluating generalist medical AI's capability to perform various tasks and interpret multimodal data. Our evaluation goes beyond accuracy to assess the impact of data modalities, compare single-task and multitask learning, and perform bias evaluation. Additionally, because it is common for real-world EHR to not have all the data modalities used for training, we also evaluate robustness against missing modalities during inference.
> In summary, our more realistic ED dataset and expanded evaluation better equip our benchmark to assess the capabilities of generalist medical AI versus existing ED/ICU benchmarks.
>
> - We would also like to clarify that our dataset will be released within a year of the submission.

---

### Decision · Program_Chairs · 2023-09-22

**Decision:**

Accept (Poster)

**Comment:**

The authors propose a Multimodal Clinical Benchmark for Emergency Care (MC-BEC). The authors provide performance baselines for each prediction task to enable the evaluation of multimodal, multi-task models.
Overall the reviewer's evaluation is positive, although some limitations in the demographic information and details about the evaluation.